# An Impact Asymmetry Analysis of Small Urban Green Space Attributes to Enhance Visitor Satisfaction

**DOI:** 10.3390/ijerph19052922

**Published:** 2022-03-02

**Authors:** Pengwei Wang, Lirong Han, Rong Mei

**Affiliations:** 1School of Tourism, Shanghai Normal University, Shanghai 200234, China; 2School of Geography and Tourism, Hulunbeier College, Hulunbuir 021008, China; ldhlr@hlbec.edu.cn (L.H.); dmr@hlbec.edu.cn (R.M.)

**Keywords:** impact asymmetry analysis, nonlinear effect, park features, small urban green spaces

## Abstract

Urban green spaces have beneficial effects on the health and well-being of citizens. Understanding the factors influencing visitor satisfaction with urban green spaces contributes to making more informed policies. Prior studies on green spaces satisfaction primarily focused on the linear correlation between small urban green space attributes and satisfaction. In this manuscript, we presented a study aimed to (1) identify the attributes of SUGS as frustrators, dissatisfiers, hybrids, satisfiers, and delighters; (2) prioritize attributes for effective satisfaction management; (3) assist managers in drafting guidelines for operational management decisions. We gathered a range of information about the users to nine SUGS in Shanghai, in 2020, via a questionnaire, and we found that safety, noise, and social interaction are improvement priorities. Squares and visitors’ behavior should not be ignored in SUGS management. Moreover, managers should carefully monitor SUGS attributes of the social environment to meet users’ expectations. The findings of this study have implications for SUGS management and future research.

## 1. Introduction

Within 30 years, 70% of the world’s population will live in cities [1]. The population migration to cities will exert a crucial impact on human health because more people will be exposed to chronic and mental diseases such as obesity, osteoporosis, heart disease, depression, and mental fatigue. The COVID-19 pandemic, followed by less outdoor physical exercise and limited social contact, has aggravated these diseases [2]. Nature exposure contributes to decreasing residents’ chronic and mental diseases [3,4,5]. Urban green space is a convenient way for urban residents to interact with nature. Urban green spaces with good location and convenient transportation can increase residents’ use [6,7]. Nevertheless, it is challenging to have more space in densely populated and well-located areas to construct large urban green spaces for cities with little land [8]. Small urban green spaces (SUGS) can effectively solve this problem. SUGS with natural characteristics will encourage people to use outdoor areas, thereby increasing social integration and interaction between people and improving residents’ well-being and health level [9,10,11,12].

Compared with large urban green spaces, SUGS play different roles in urban management and residents’ lives, which leads to different park design and use patterns [13], such as use frequency and motivations [14,15]. Visitors primarily use SUGS mainly for functional needs rather than for aesthetics, such as rest and restitution and socializing, and older adults are more likely to be frequent users of SUGS. The simple natural environment in SUGS is not necessarily beneficial or attractive to people [16]. Evidence indicates that the need of visitors for various types of park attributes is different, and exploring the correlation between SUGS attributes and satisfaction is a crucial means to augment the attractiveness of parks and at the same time increase the well-being of residents.

Prior studies on urban green spaces satisfaction primarily focused on the linear correlation between urban green space attributes and overall green space satisfaction. Nevertheless, visitors’ satisfaction with green space is the result of the interaction among multiple factors. The enhancement of certain urban green space attributes would increase visitor satisfaction, while the lack of certain attributes might cause visitor dissatisfaction. The asymmetric relationship describes the phenomenon that the performance of quality attributes exerts different influences on the satisfaction of asymmetry effects [17,18,19]. Ignoring the dynamic link between quality attributes and satisfaction could lead to improper model specification and lower predictive power [20]. The asymmetric relationship between service attributes and overall satisfaction has been well recognized in the satisfaction literature [21]; it has also been used in some studies on urban environments [22]. However, none explores the asymmetric relationship between SUGS attributes and visitor satisfaction.

Based on the abovementioned studies, we use nine SUGS in Shanghai to explore if an asymmetric relationship exists between SUGS attributes and visitor satisfaction using impact asymmetry analysis (IAA). The study aims to (1) identify the attributes of SUGS as frustrators, dissatisfiers, hybrids, satisfiers, and delighters; (2) prioritize attributes for effective satisfaction management; (3) assist managers in drafting guidelines for operational management decisions.

## 2. Literature Review

### 2.1. The Asymmetric Effect of Attributes on Satisfaction

To date, many studies have examined the symmetric relationship between attributes and satisfaction but ignored that the asymmetric impact of qualitative attributes on satisfaction limits insight into attributes more sensitive to satisfaction or dissatisfaction. The asymmetry function is expressed as positive and negative asymmetry [17]. While positive asymmetry implies that an attribute is more sensitive to satisfaction than to dissatisfaction, negative asymmetry implies that an attribute produces more dissatisfaction than satisfaction [18,19,20]).

The aforementioned dynamic effect of attributes on overall satisfaction has been explored using the three-factor theory of satisfaction [21,23], which is based on the two-factor theory by Herzberg et al. (1959) that claimed that the factors that lead to job dissatisfaction differ from those that lead to job satisfaction (such as challenging work) [24]. Kano (1984) proposed an attractive quality theory, and it is based on five quality dimensions that affect satisfaction differently [25]. The five quality dimensions are categorized into attractive quality, must-be quality, one-dimensional quality, indifferent quality, and reverse quality. Attractive quality denotes the value-added attributes that users do not usually expect. When attractive attributes are given, the user is satisfied. However, even when it is not available, users will not be dissatisfied or disappointed. Thus, the attractive quality is an asymmetric attribute. Compared with attractive quality, must-be quality is regarded as a basic attribute. When this attribute is not provided or fails to fulfill their expectations, users are likely to be dissatisfied. However, even when it fulfills their requirements and expectations, they might still be dissatisfied because users take the must-be attribute for granted. Hence, the must-be attribute is considered a negative asymmetric attribute. One-dimensional quality affects both positive and negative user satisfaction [25]. Alternatively, if the attributes are not provided, it would affect both the positive and negative satisfaction of users. Indifference quality is that whether or not it is provided, it does not affect satisfaction or dissatisfaction. Reverse quality means that if the attribute is provided, users may not be satisfied.

Then, some studies adjusted Kano’s (1984) attractive quality theory into the three-factor structure of attributes that cause satisfaction and/or dissatisfaction [19,21,23,25]. For example, Oliver (1997) suggested that satisfaction is differently influenced by three types of attributes—bivalent satisfiers, monovalent dissatisfiers, and monovalent satisfiers [19]. Lee et al. (2017) explored the asymmetric nature of attributes from three zones, including negative asymmetry, hybrids attributes, and positive asymmetry [26]. Negative asymmetry involves dissatisfiers and frustrators. Hybrid attributes exhibit positive symmetric linear correlations with satisfaction. Positive asymmetry comprises satisfiers and delighters. As this classification method has been extensively used in many studies [21,27], it was adopted in this study.

Multiple studies have investigated the asymmetry of many attributes affecting customer satisfaction, including incentive travel [26], ski resorts [28], restaurants [23], casinos [27], urban tourism [29], neighborhood and communities [22], aviation services [30], and national forest park [31].

### 2.2. Resident Perception and Satisfaction with Green Space Attributes

The previous literature suggests that people’s perception and satisfaction with green space are affected by the spatial and environmental characteristics of green space [32], the existence and quality of facilities [33,34,35], management or maintenance [36], the behavior of other users [33,37], and socioeconomic factors [38].

Regarding spatial and environmental characteristics, some studies reported that the location, size, and environment of the park (e.g., shadows along the road, walking paths, water features, and birds) affect residents’ perception and use of the park [39]. Rey Gozalo et al., (2019) explored the impact of 12 park attributes, such as green space, aesthetics, location, noise, and air, on the satisfaction of large and small green space parks and reported that visitors’ overall satisfaction with both types of parks correlated with noise, satisfaction with large parks correlated with park conservation, and satisfaction with small parks correlated with safety and users [40]. This indicates that providing clean, safe green space is essential for visitors. Rey Gozalo et al., (2018) established that residents’ satisfaction with noise significantly affected the overall satisfaction of green spaces [41]. Residents’ satisfaction with the characteristics of green spaces can justify 71.4% of the overall satisfaction with green spaces. Research on structural preferences indicates that the public is more inclined to larger trees and older forest structures [42]. The perception of trees can be very subtle and even depend on tree parameters [43], and a few studies specify that tree species are not a crucial factor in human satisfaction [33].

In addition, satisfaction with green parks depends on the availability and status of facilities that meet different user groups [34,35]). Older people might need benches, parents with small children are more likely to need play facilities, cyclists will choose parks with bike lanes green spaces [5], and the presence of sports fields or outdoor fitness equipment could increase youth satisfaction with parks. Convenient facilities and higher usability can help augment the satisfaction of the park [44]. Elements such as walking trails, barbecues, picnic tables, public restrooms, and lighting contribute to visitor satisfaction in country parks; having insufficient activity facilities is considered one of the factors limiting the use of SUGS [45].

Some previous studies illustrated the significance of management and maintenance to uphold the attractiveness of green parks [36]. Jim and Chen (2006) established that poor maintenance weakened the attractiveness of urban green space and caused negative perceptions [38]. Trash on trails could exert a negative impact on people’s overall entertainment experience [46], and signs of negligence and vandalism are one of the key reasons for fear of crime [47]. Moreover, the provision of park regulations [48] and the number of organized activities [49] will affect residents’ perception and satisfaction of the park. Roberts et al. (2019) reported that the misbehavior of other visitors influenced visitors’ satisfaction [44]. User misbehavior is manifested as littering [46], loud noises, vandalism, and dog pollution [33]. In addition, the fact that dogs are not on a leash might hinder the appearance of other user groups, including children [37]. Furthermore, people who like quiet places avoid crowded and noisy areas [35].

Studies on SUGS have focused on SUGS use, benefits, and satisfaction, for instance, the relationship between SUGS and human well-being, psychological recovery [11], use motivations, use patterns, and the correlation between SUGS attributes such as aesthetics, noise, safety, and visitor satisfaction [14,40,45].

The existing studies explored the correlation between green space attributes and satisfaction from the aspects of spatial and environmental attributes, facilities, management, and behavior of other visitors. They usually focused on the linear relationship between green space attributes and visitor satisfaction, making it impossible to reveal the potentially complex nonlinear effect of attributes on visitor satisfaction, which might result in biases [50]. Hence, this study takes SUGS as the research object and fills a gap in the literature by introducing the impact asymmetry analysis developed in management science into the research of SUGS satisfaction and reports the complex effects of SUGS attributes on visitor satisfaction. The study contributes to prioritizing attributes for effective satisfaction management and improving visitors’ satisfaction in a cost-effective manner.

## 3. Study Methods

### 3.1. Study Site

Shanghai is in the Western Pacific, east of the Asian mainland. With a total area of 6340.5 km^2^, Shanghai is China’s largest city and a highly international city. In 2020, the total population of Shanghai was 24.87 million, and the per-capita disposable income was 72,232 RMB, ranking first in China. By the end of 2020, the per-capita green area of parks in Shanghai reached 8.5 m^2^.

In the next 5 years, Shanghai will focus on building a park city, a forest city, and a wetland city under the guidance of the goal of building an ecological city. The 14th five-year plan proposed that Shanghai will strive to promote the continuous enhancement of residents’ living environment and augment the residents’ sense of gain and satisfaction. SUGS will play a crucial role for Shanghai to construct an ecological city and improve residents’ satisfaction and happiness.

Shanghai has 16 districts, including 7 central districts, which are densely populated areas. There are 32 SUGS with a size of less than 2 ha each in 7 central districts. This study selected 9 SUGS located in 7 central districts. Each park is easily accessible, surrounded by dense residential areas, and these parks are open to the public free of charge every day. Table 1 shows the details of the nine SUGS.

### 3.2. Data Collection

A questionnaire method was used to collect data. There were three parts in the questionnaire: the first part enquired about the usability pattern of park visitors, including use motivation, activities, stay time, use frequency; the second part was about visitors’ overall satisfaction and satisfaction with park attributes; the third part collected the social and demographic data of SUGS visitors.

We conducted a pilot test with 40 park visitors before the formal survey. A formal survey was conducted in nine SUGS from August 2020 to November 2020. Six university students conducted a face-to-face survey after being trained. We distributed 600 formal questionnaires and collected 507 valid questionnaires. We informed all the respondents of the confidentiality of the research and explained our privacy plan. Respondents were randomly selected from those who visited SUGS. The first respondents were randomly selected. Then, the fifth visitor who passed next was invited to take part in the survey. If the invited visitor refused, we repeated the procedure [15,51].

### 3.3. Measurement Development

Based on the project development procedures used by Churchill (1979) and DeVellis (1991), the measurement methods used in this research were determined and developed through literature review and in-depth interviews [52,53]. In total, 14 attributes were initially extracted from the literature review, which included the number and diversity of tree species, the area and quality of green land (green land), the distance from the park to the residential area (distance), safety, the size and quality of squares (squares), aesthetics, design of walking paths, noise, amenities, leisure facilities, and sports facilities, the organization of park activities, park maintenance, and other visitors’ behavior. Our previous investigation revealed that the interaction among visitors was frequent, and there were many group activities in SUGS. Thus, in accordance with the study by Dong et al. 2019, two variables were added—ease of seeing friends and social interaction [22]. The overall satisfaction of visitors and the 16 attributes mentioned above were measured using the five-point Likert Scale method. Finally, six experts were invited to determine the design rationality.

### 3.4. Exploratory Factor Analysis

We used the Kaiser-Meyer-Olkin (KMO) test and Bartlett’s test to assess the rationality of the factor analysis for the 16 attributes. The KMO test ensured that the overall measure of sampling adequacy was 0.832 (>0.80), and the *p* value of Bartlett’s test was <0.001, which supported the rationality of the factor analysis. The attributes were removed when they were below the factor loading value of 0.4 and communality of 0.5. The diversity of tree species and aesthetics attributes were loaded <0.40 and hence were removed. The remaining 14 attributes were classified into the following four dimensions: spatial and environmental characteristics (5 attributes), social environment (4 attributes), presence and quality of facilities (3 attributes), and park management or maintenance (2 attributes).

### 3.5. Reliability, Construct Validity, and Method Biases

The AVE values in Table 2 are 0.5 or greater, which supports convergence validity (Fornell and Larcker, 1981) [54]. Cronbach’s value of 0.70 has been widely accepted as the cutoff point for measuring the reliability of the scale [29]. The reliability of the park management or maintenance dimension does not exceed the cutoff point of 0.70, and therefore, the two attributes in this dimension were removed. Finally, a total of 12 attributes were used that focused on the following 3 dimensions (Table 2): spatial and environmental characteristics (5 attributes), social environment (4 attributes), presence and quality of facilities (3 attributes), explaining 63.7% of the data variance.

In addition, as suggested by Schriesheim (1979) and Podsakoff et al. (1984), we used the univariate analysis to check common methods bias and examined the data via principal component analysis and varimax rotation [55,56]. If a single factor structure was identified or one factor explained >50% of the variance, it provided evidence of a common method bias. The univariate analysis explained 23.8% of the variance using the first dimension, suggesting that the common method bias was a negligible issue in this study. Furthermore, this study tested for collinearity before running the models. We calculated the former’s variance inflation factors, and all were found to be <10 [30].

We checked variance homoscedasticity and the normal distribution. No significant difference was found in the variance of each attribute (*p* > 0.05). The absolute values of skewness of all attributes were <3, and the absolute values of kurtosis of all attributes were <10, showing that the data did not deviate significantly from the normal distribution based on the standard of Kline (1998) and Wang et al. (2021) [57,58]. Table 2 presents the skewness and kurtosis of each attribute.

### 3.6. Impact Range Performance Analysis (IRPA) and IAA

We used impact range performance analysis (IRPA) and IAA to determine the asymmetric effects of attributes on satisfaction [21]. Penalty–reward contrast analysis (PRCA) was carried out using multiple regression analysis and dummy variables to estimate IRPA and IAA [59]. Specifically, two sets of dummy variables must be generated for each attribute as follows:(1)The first group of dummy variables (penalty index) was created to estimate the impact of the low performance of an attribute on satisfaction; this was performed by encoding the lowest score of an attribute as 1. When the representation of the attribute was 1, it was entered as 1. Zero was entered for attribute scores 2, 3, 4, and 5.(2)To evaluate the effect of the high performance of an attribute on satisfaction, the second set of reward indices was generated by encoding the highest score of an attribute as 1. When an attribute had a score of 5, 1 was entered. Zero was entered for attribute scores 1, 2, 3, and 4.(3)Then, satisfaction regression was conducted for the two dummy variables, and penalty and reward indices were obtained. Penalty indices (PIs) indicate the properties that negatively correlate with satisfaction, while Reward indices (RIs) indicate the properties that positively correlate with satisfaction. Table 3 shows PI and RI values.

After the estimation of PI and RI, the value of an attribute’s range of impact on satisfaction was estimated (RIS). The absolute values of PI and RI of each attribute were added to generate RIS (Table 3), indicating the degree of influence of attributes on satisfaction. Next, PI, RI, and RIS were introduced into the equations developed by Mikulić and Prebeẑac (2008) to create satisfaction-generating potential (SGP) and dissatisfaction-generating potential (DGP) [21]. The SGP and DGP depict the rates of penalty and reward; RIS values are used to estimate the impact asymmetry (IA) as follows:(1)SGPi = RI/RISi(2)DGPi = |PI|/RISi(3)IAi = SGPi − DGPi
where RI = reward index for attribute *i*; PI = penalty index for attribute *i*; RISi = |PI| + RI = range of impact on attribute score; SGPi + DGPi = 1.

As IA is measured according to the arithmetic difference between SGP and DGP, IA acts as a cutoff point in classifying attributes as dissatisfying, mixed, or satisfying. In particular, if an attribute’s SGP is higher than the corresponding DGP, the attribute is considered more satisfactory than unsatisfactory, and the attribute is classified as satisfier. Conversely, when the DGP is greater than the response SGP, the property is classified as dissatisfier. In addition, when the arithmetic difference between the SGP and DGP is marginal for an attribute, the attribute is categorized as a hybrid because hybrid exerts a comparable effect on satisfaction and dissatisfaction.

Using the thresholds of Back and Lee (2015) and Dong et al. (2019) [22,27], we grouped the attribute into one of the following five factors: a frustrator if IA < −0.7; a dissatisfier if −0.7 ≤ IA< −0.2; a hybrid if −0.2 ≤ IA ≤ 0.2; a satisfier if 0.2 < IA ≤ 0.7; a delighter if IA > 0.7.

Dissatisfiers and frustrators exhibit negative asymmetric effects. Dissatisfiers cause dissatisfaction if they perform poorly. Frustrators are regarded as severe dissatisfiers. Dissatisfiers and frustrators exert a limited impact on the overall satisfaction once they perform well. Hence, planners must improve them only if they do not perform well and only improve them to a level that fulfills visitors’ expectations to minimize overinvestment.

Hybrids exhibit positive symmetric linear correlations with satisfaction. If they perform well, people are satisfied, but dissatisfaction occurs when they perform poorly; the priority level of these attributes is lower than that of frustrators/dissatisfiers.

Satisfiers are deemed value-added properties because they offer more value and pleasant surprises. Delighters are considered as a high level of satisfiers. When dissatisfiers, frustrators, and hybrids perform well, more investment can be made in value-added properties.

## 4. Results

### 4.1. Descriptive Statistics and Usability Pattern

Table 4 shows the socioeconomic characteristics and behavior patterns of visitors. A small difference existed in the proportion of male and female visitors, and the main age of visitors was 21–30, 31–40, and >60 years. Most of the visitors were married, and the main degree appeared to be a bachelor’s degree. The average income of the visitors was low, and the majority of visitors’ income concentrated between 3000 and 6000 RMB, which was lower than the average income level of Shanghai in 2019 (114,962 RMB/year, Source: Shanghai Statistical Yearbook for 2020). The reasons for the above differences need further study.

The duration of stay is usually 1–2 and 2–4 h. The types of visitor activities include individual and small group activities (2–4 people). Small-group activities primarily include games such as poker and chess and also chat-based activities. The frequency of visits is around 1–3 times a month and 1–2 times a week. The majority of visitors reported using the park for relaxation and rest (52.7%), corroborating the findings from other studies [9,14]. Physical exercise is also one of the purposes for residents to visit SUGS. SUGS is a crucial resource for physical activity and a catalyst for promoting physical activity among residents because they are free, inexpensive, and accessible to most residents [60]. A total of 17.2% and 16.2% of visitors used SUGS for a walk and to take their children out to play, respectively. Only 3.7% of visitors went to SUGS for aesthetics. Results on visitors’ motivation suggested visitors’ use of SUGS largely for functional needs (relaxation and rest, physical exercise, walking, and taking children out) rather than for aesthetics.

### 4.2. The Impact of Descriptive Statistics and Usability Pattern on Satisfaction

The results of independent sample *t*-test and one-way ANOVA showed that visitors with different ages, educational background, gender, income, marriage, stay time, visit motivation and frequency had no difference in overall satisfaction, while visitors participating in different activities have significant differences in the park overall satisfaction (*p* < 0.05). Multiple comparisons revealed that the visitors who participated in individual activities were less satisfied with SUGS than the visitors who participate in small group activities (2–4 people) and group activities (≥5 people; Figure 1).

### 4.3. Differences and Similarities on Attributes of Nine SUGS

This study analyzed the differences between the 12 attributes of 9 SUGS. One-way ANOVA showed that visitors’ satisfaction with noise in nine parks differed significantly (*p* < 0.05). Multiple comparisons revealed that visitors in Yanchun, Hudiewan, and Jiaotong parks were less satisfied with the noise than the visitors in Liyuan, Yichuan, and Liangcheng Parks (Figure 2), primarily because Yanchun, Hudiewan, and Jiaotong Parks are next to the subway, and the subway is the main noise source of the parks.

There was no significant difference in overall satisfaction and other attributes’ satisfaction of the nine SUGS; this could be because, although some differences exist in internal design and attributes of nine SUGS, the functions provided by these attributes are similar. For example, most SUGS paths are stone roads and gravel roads, and a few SUGS have plastic paths. The paths in SUGS are relatively curved and narrow, and their width is about 1–2 m. They are suitable for walking and partially suitable for wheelchair users. However, they cannot be used for other activities, such as skateboarding and roller skating. SUGS are equipped with similar fitness facilities, amenities, and leisure facilities, and only sports facilities are slightly different. For instance, some SUGS are equipped with sports facilities such as shared basketball courts and badminton courts.

### 4.4. Results of IRPA and IAA

Based on IRPA and IAA, this study evaluated the asymmetrical effect of each attribute of SUGS on visitor satisfaction from three dimensions using 12 attributes and created a coordinate chart of service quality (Figure 3), in which the *X*-axis corresponds to RIS, and the *Y*-axis corresponds to IA. The *X*-axis has two dividing lines; the first is based on the average value between the minimum and the average RIS value, while the second is based on the average value between the maximum and the average RIS value, dividing the chart into three regions, corresponding to low impact, moderate impact, and high impact [31]. Per the four boundaries of Back and Lee (2015) [27] and Dong et al., (2019) [22], the *Y*-axis divides the graph into five regions—delighters, satisfiers, hybrids, dissatisfiers, and frustrators.

Figure 3 shows the results. The results verified the asymmetric effect of quality attributes. Among the 12 attributes, three attributes—squares, walking paths, and other visitors’ behavior—had linear correlations with visitors’ satisfaction, while the other attributes had nonlinear correlations. Amenities and safety were high-impact frustrators, whereas leisure facilities were moderate-impact frustrators, and noise, green land, and social interaction were low-impact frustrators; distance was a moderate-impact dissatisfier, whereas sports facilities and easily meeting friends were moderate-impact delighters.

## 5. Discussion and Conclusions

### 5.1. Theoretical Contributions

This study provided empirical evidence of the correlation between SUGS attributes and visitors’ satisfaction and assesses the attributes that would lead to visitors’ satisfaction or dissatisfaction. In addition, this study identified improvement priorities among the attributes of SUGS and assisted managers in formulating guidelines for operational management decisions. This study is conducive to promoting the green city program launched by OECD, improving environmental quality and sustainable development of metropolitan areas, thus enhancing the contribution of urban areas to people’s quality of life and social welfare.

The main conclusions can be summarized as follows: (1) This study classified SUGS attributes into frustrators, dissatisfiers, hybrids, satisfiers, and delighters; (2) safety, noise, and social interaction are improvement priorities; squares and visitors’ behavior should not be ignored in SUGS management; (3) managers should carefully monitor SUGS attributes of the social environment to meet users’ expectations.

Regarding theoretical contributions, although some studies focused on the relationship between park characteristics and satisfaction, most studies only focused on the linear relationship (or symmetric effect). When park attributes scores are higher, the likelihood of overall satisfaction increases. This study introduced impact asymmetry analysis into the field of SUGS, a framework developed in management science. This method relaxes the assumption that the performance of service attributes exerts a linear impact on the overall satisfaction [61], and it divides SUGS attributes into more detailed categories than importance–performance analysis. In addition, this study demonstrated that SUGS attributes exert asymmetrical effects on satisfaction and dissatisfaction and play different roles, thereby validating the applicability of Herzberg’s two-factor theory and Kano’s three-factor theory to SUGS satisfaction. Overall, the attributes identified by our study can provide good points of reference for future SUGS research.

### 5.2. Management Implications

SUGS managers should carefully monitor frustrators and dissatisfiers with low attribute performance scores, such as noise, safety, and social interaction (Table 3). Safety with higher RIS scores is an improvement priority, followed by noise and social interaction.

Following safety risks still exist in SUGS. First, there is occasional theft and other behaviors, such as visitors hanging their clothes on the hanging place in the process of sports and leisure, resulting in the loss of property. Second, some parks have large areas of water; however, there is no fence, which presents a safety risk to children. Low-density shrubs positively affect personal security. When designing parks, attention should be paid to the type and density of vegetation. Furthermore, improving lighting, fences, surveillance, lockers, and other elements can create a safe park environment.

In urban management, quietness is a priority of the World Health Organization [62]. Rey Gozalo and Barrigón Morillas (2017) mentioned that the disappearance of noise is the most influential environmental feature determining the overall perception of an urban area [63]. The main sources of noise in green parks are road traffic, children playing and screaming, dance music, and construction sites, among which road traffic is the main source of noise in SUGS. As regards site selection, SUGS sites should be selected considering the function of nearby urban roads [64]. In addition, noises related to children playing and screaming, as well as dance music, should also be given adequate attention. The purpose of visits is different among visitors, and the demand for the park environment is different; many visitors do not like noisy music, while others dislike children’s frolicking. Visitors tend to interact with people of the same age group, and it is difficult for them to interact across age groups or classes [65]. Thus, through reasonable space design, the corresponding management measures should be introduced, and the decibel level of music in the park should be stipulated to decrease the mutual interference between visitors.

Poor social interaction will lead to visitors’ dissatisfaction. In addition, we found that visitors who participate in individual activities are less satisfied with SUGS than the visitors who participate in group activities. The main ways for visitors to interact are playing poker, chess, badminton, and dancing. Interaction between visitors often occurs between like-minded peers and peers, and people prefer to interact with “familiar or familiar faces” for passive selective interaction. The park can augment the interaction between visitors by creating interest societies and organizing park activities, provide more social participation and communication opportunities for visitors, especially the elderly, increase community cohesion, and improve the physical and mental health of urban residents, thus making a wider contribution to the well-being of the surrounding residents [65].

In addition, SUGS managers should not overlook squares and visitors’ behavior with low-attribute performance scores, which were categorized as hybrids, especially squares with relatively high RIS.

Squares are used for performing various activities, such as dancing and tai chi, which are popular among middle-aged and elderly people. In this study, Liangcheng Park, Yichuan Park, and some other parks have their small squares, but the squares are basically open-air, without a canopy or sunshade. Furthermore, attention should be paid to layout and functional zoning in the design of SUGS Square. Dancing music and children playing in the squares are the main sources of noise.

This study demonstrated that the mean scores of visitors’ behavior were only 3.25. The misbehavior of visitors is an urgent problem for SUGS to resolve. Visitors’ misbehavior includes littering, spitting, and destroying vegetation. Visitor misbehavior can be decreased by intensifying patrols, increasing cameras, increasing publicity, and education.

This study revealed that the attributes of the social environment are all with low attribute performance scores. Most attributes related to the social environment were categorized as either hybrid or frustrator. Visitors would be easily dissatisfied when confronted with social environment issues. Thus, managers should carefully monitor SUGS attributes of the social environment to meet users’ expectations. Compared with large parks, in SUGS, a good social environment is a unique expectation of visitors. Shanghai SUGS play the role of community parks to some extent. Wang et al. (2021) reported that older adults were more likely to be frequent users of SUGS [15]. The WHO (2007) stated that social participation and social inclusion are the major domain for attaining age-friendly cities [66]. Managers should consider how to make SUGS play greater roles in promoting social participation and designing cities that are friendly and supportive to aging populations.

### 5.3. Limitations

This study has several limitations. First, we obtained the measurement items from previous studies. In future studies, we can introduce the mixed-method and define measurement items by visitor interviews. Second, the distribution of attributes depends on the location of crosshairs. According to the definition of cutoff points [22,27], AIPA divides the attributes into five types. Changing the cutoff points will lead to a change in assignment. Third, the study relied on cross-sectional data and self-reported measures. Carryover effects cannot be considered when cross-sectional data are used. As cross-sectional studies can only explain phenomena at a particular point in time, the results of cross-sectional studies differ if different time periods are selected. Forth, the research samples are taken from SUGS in Shanghai; hence, whether these research results can be applied to other cities in China warrants further investigation. To complement the lack of generalization, more research needs to be carried out in other Chinese cities.

## Figures and Tables

**Figure 1 ijerph-19-02922-f001:**
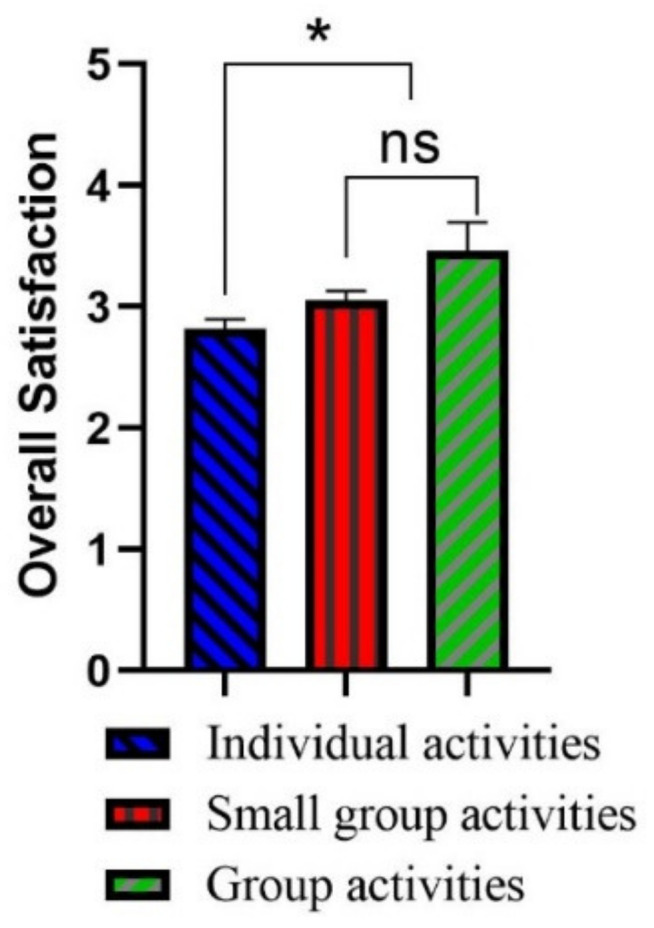
The impact of different activities on satisfaction. * Significant at *p* < 0.05. ‘ns’ indicates no significant difference.

**Figure 2 ijerph-19-02922-f002:**
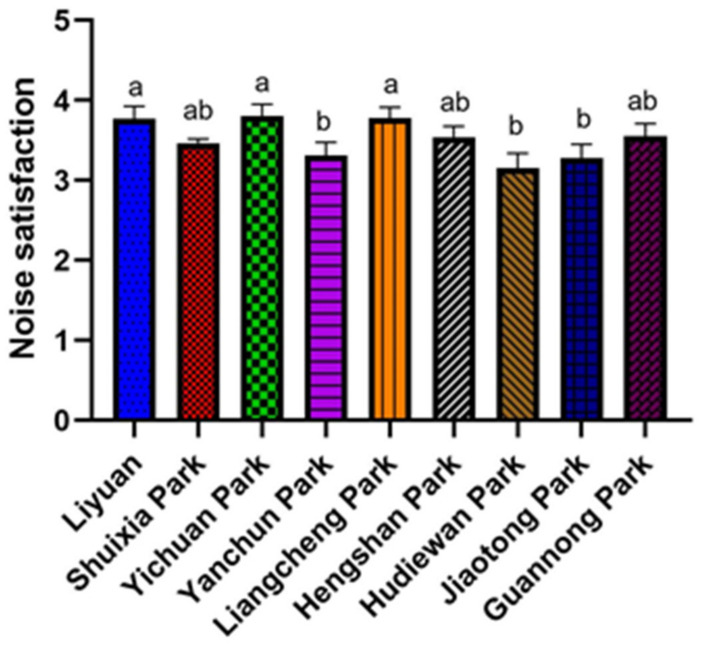
Differences in noise satisfaction among nine SUGS. Different letters indicate the difference between groups is significant at 0.05 level.

**Figure 3 ijerph-19-02922-f003:**
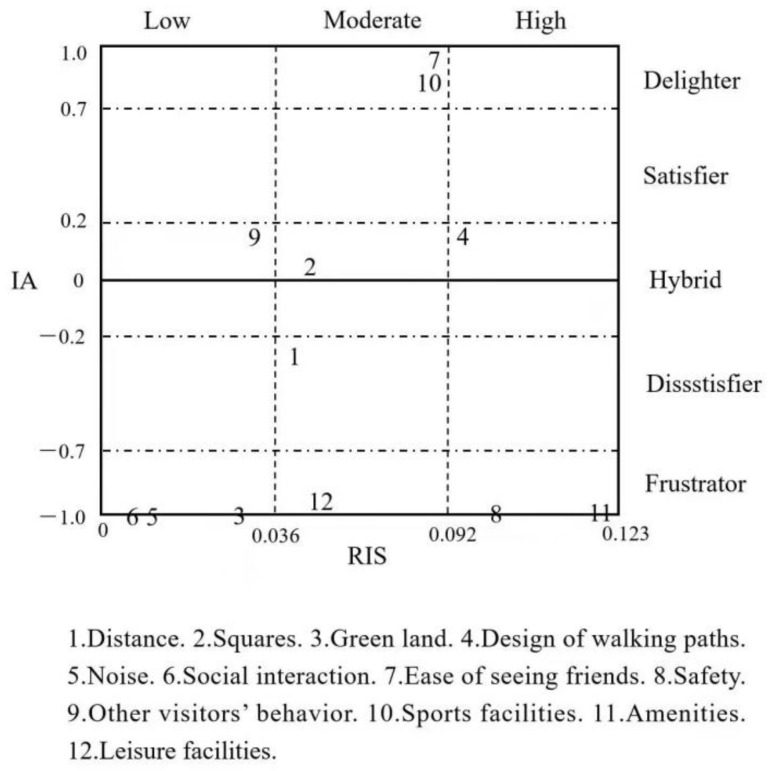
IRPA grid.

**Table 1 ijerph-19-02922-t001:** The details of the nine SUGS.

Name	Size	Location	Sport and Entertainment Facilities	Natural Conditions
Liyuan Park	1.7 acres	Liyuan Road, Huangpu District	Fitness facilities, shared basketball courts, and activity square	Trees, bushes, a few flowers, and an empty green space
Yanchun Park	1.29 acres	Yingkou Road, Yangpu District	Fitness facilities and slide (for children)	Trees, bushes, and a few flowers
Jiaotong Park	1.58 acres	Xinma Road, Jingan District	Fitness facilities, shared basketball courts, and slide (for children)	Trees, bushes, and a few flowers
Hengshan Park	1.19 acres	Guanyuan Road, Xuhui District	Fitness facilities, badminton courts, and activity square	Trees, bushes, a few flowers, and an empty green space
Yichuan Park	1.88 acres	Yichuan Road, Putuo District	Fitness facilities, slide (for children), and activity square	Trees, bushes, a few flowers, and a lake
Guannong Park	1.25 acres	Guannong Road, Putuo District	Fitness facilities and slide (for children)	Trees, bushes, and a few flowers
Hudiewan Park	1.6 acres	Kangding East Road, Jingan District	Shared basketball court and activity square	Trees, bushes, a few flowers, and a river
Shuixia Park	1.18 acres	Xianxia Road, Changning District	Fitness facilities and slide (for children)	Trees, bushes, a few flowers, and a stream
Liangcheng Park	1.37 acres	Chezhan North Road, Hongkou District	Fitness facilities, slide (for children), and activity square	Trees, bushes, and some flowers

**Table 2 ijerph-19-02922-t002:** Results of exploratory factor analysis, Skewness, and Kurtosis of all attributes.

Factors	Skewness	Kurtosis	Factor Loading	Cronbach *α*	AVE	CR
Factor 1: Spatial and environmental characteristics (% of variance: 23.8)						
Distance	−0.771	−0.538	0.654	0.809	0.51	0.84
Design of walking paths	−0.750	−0.582	0.737			
Noise	0.444	0.706	0.756			
Green land	0.701	0.728	0.727			
Squares	0.355	0.776	0.702			
Factor 2: Social environment (% of variance: 21.3)						
Safety	−0.383	−0.856	0.843	0.798	0.57	0.84
Social interaction	−0.270	−0.784	0.684			
Ease of seeing friends	−0.460	−0.763	0.857			
Other visitors’ behaviors	−0.245	−0.742	0.600			
Factor 3: Presence and quality of facilities (% of variance: 18.6)						
Sports facilities	−0.595	−0.629	0.727	0.815	0.64	0.84
Leisure facilities	−0.707	−0.451	0.853			
Amenities	−0.786	−0.086	0.823			
Over satisfaction	0.222	−0.804				

**Table 3 ijerph-19-02922-t003:** Results of IRPA and IAA.

Dimensions	Attributes	RI	PI	RIS	SGP	DGP	IA	Type	Mean Scores
Spatial and environmental characteristics	Distance	0.016	**0.034**	0.05	0.32	0.68	−0.36	Dissatisfier	3.79
	Squares	0.028	0.024	0.052	0.54	0.46	0.08	Hybrid	3.49
	Green land	−0.017	0.012	0.029	−0.59	0.41	−1.00	Frustrator	3.71
	Design of walking paths	**0.058**	**0.043**	0.101	0.57	0.43	0.15	Hybrid	3.72
	Noise	−0.008	0.003	0.011	−0.73	0.27	−1.00	Frustrator	3.51
Social environment	Social interaction	−0.009	0.001	0.01	−0.90	0.10	−1.00	Frustrator	3.23
	Ease of seeing friends	**0.082**	0.001	0.083	0.99	0.01	0.98	Delighter	3.46
	Safety	**−** **0.103**	0.001	0.104	−0.99	0.01	−1.00	Frustrator	3.37
	Other visitors’ behaviors	0.018	0.012	0.031	0.58	0.39	0.19	Hybrid	3.25
Presence and quality of facilities	Sports facilities	**0.085**	0.005	0.09	0.94	0.06	0.89	Delighter	3.71
	Amenities	−0.016	**0.107**	0.123	−0.13	0.87	−1.00	Frustrator	4.00
	Leisure facilities	0.004	**0.049**	0.053	0.08	0.92	−0.85	Frustrator	3.86

Bold values *p* < 0.05. RI: reward index; PI: penalty index; RIS: range of impact on customer satisfaction (|PI| + |RI|); SGP: satisfaction-generating potential (RI/RISi); DGP: dissatisfaction-generating potential (|PI|/RISi); IA: impact asymmetry (SGPi − DGPi).

**Table 4 ijerph-19-02922-t004:** Demographics of the visitors who participated in the study.

Variable		Total Sample	Percentage	Variable		Total Sample	Percentage
		Number				Number	
Gender	Male	245	48.3%	Stay time	<30 min	15	3%
	Female	262	51.7%		30 min–1 h	120	23.7%
Age	18–20	36	7.1%		1 h–2 h	179	35.3%
	21–30	135	26.6%		2 h–4 h	147	29%
	31–40	126	24.9%		4 h–6 h	39	7.7%
	41–50	58	11.4%		>6h	7	1.4%
	51–60	43	8.5%	Activities	Individual activities	225	44.4%
	>60	109	21.6%		Small group activities (2–4 people)	254	50.1%
Education	High school and below	142	28%		Group activities (≥5 people)	28	5.5%
	College	94	18.5%	Frequency	1–3 times a month	284	56%
	Undergraduate	207	40.8%		1–2 times a week	106	20.9%
	Postgraduate or above	64	12.6%		3–5 times a week	51	10.1%
Monthly Income(RMB)	<3000	77	15.2%		every day	66	13%
	3000–6000	147	29%	Motivation	Relaxation and rest	267	52.7%
	6000–8000	113	22.3%		Physical exercise	136	26.8%
	8000–10,000	86	17%		Walk	87	17.2%
	>10,000	84	16.6%		Take children out	82	16.2%
Marriage	Single	150	29.6%		Meet with friends	44	8.7%
	Married	348	68.6%		Access to nature and fresh air	40	7.9%
	Divorced	9	1.8%		Enjoy the aesthetics	19	3.7%
					Other	13	2.6%

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
