# Peer review of "An Impact Asymmetry Analysis of Small Urban Green Space Attributes to Enhance Visitor Satisfaction"

_ijerph, 2022, doi:10.3390/ijerph19052922_

Round 1

Reviewer 1 Report

It is a very good work, with a very consistent theoretical framework, methodology and analysis of results. Didn't identified any major errors or problems. It does a good contribution on management decision making process and can be useful for future local planning of cities. Just one remark: conclusion is too brief, should enphasize better main results and development strategies.

Author Response

Response to Reviewer 1’s Comments

Manuscript: ijerph-1592446

Title: An impact asymmetry analysis of small urban green spaces attributes to enhance visitors’ satisfaction

Dear Reviewer 1:

The authors would like to thank the area reviewer for your time and comments. Your feedback is very important. We have carefully addressed each of your suggestions and given the corresponding answers.

---------------------------------------------------------------------------------------------------------------------

Reviewer’s Comments

Responses

It is a very good work, with a very consistent theoretical framework, methodology and analysis of results. Didn't identified any major errors or problems. It does a good contribution on management decision making process and can be useful for future local planning of cities.

We thank the reviewer for your positive feedback and comments.

Just one remark: conclusion is too brief, should enphasize better main results and development strategies.

Thank you for this comment.As the conclusion is too brief, we elaborated the significance of this study in more detail (please see lines 439-442). In order to avoid duplication with the discussion, we combined the conclusion and discussion (please see lines 434-447).

Again, to the reviewer 1, we thank you very much for your kind inputs. We have also learned a lot from your thoughtful comments, which have helped improve this study’s contents.

Sincerely yours,

The Authors

Reviewer 2 Report

Review on ijerph-1592446

Good paper with a significant and valuable topic.

There are too many figures that show the urban parks. Only one or two are enough. Also, only the name and size are enough for each park.

‘5. Discussion’ should be changed to ‘5. Discussion and Conclusion’.

The current ‘6. Conclusion’ has to be moved to 5.1. Discussion.

I would like to strongly suggest citing the following references.

Kim, J. S., Lee, T. J., & Hyun, S. S. (2021). Estimating the economic value of urban forest parks: Focusing on restorative experiences and environmental concerns. Journal of Destination Marketing & Management. DOI:10.1016/j.jdmm.2021.100603.

Author Response

Response to Reviewer 2’s Comments

Manuscript: ijerph-1592446

Title: An impact asymmetry analysis of small urban green spaces attributes to enhance visitors’ satisfaction

Dear Reviewer 2:

The authors would like to thank the area reviewer for your time and comments. Your feedback is very important. We have carefully addressed each of your suggestions and given the corresponding answers.

---------------------------------------------------------------------------------------------------------------------

Reviewer’s Comments

Responses

Good paper with a significant and valuable topic.

We thank the reviewer for your positive feedback and comments.

There are too many figures that show the urban parks. Only one or two are enough. Also, only the name and size are enough for each park.

Thank you for this comment.We deleted the figures in Table 1 and added the contents of the table 1.

5. Discussion’ should be changed to ‘5. Discussion and Conclusion’.The current ‘6. Conclusion’ has to be moved to 5.1. Discussion.

Thank you for this comment.We moved 6. Conclusion to  5.1 Discussion and combined the the section of conclusion and discussion (please see lines 434-447).

I would like to strongly suggest citing the following references.

Kim, J. S., Lee, T. J., & Hyun, S. S. (2021). Estimating the economic value of urban forest parks: Focusing on restorative experiences and environmental concerns. Journal of Destination Marketing & Management. DOI:10.1016/j.jdmm.2021.100603.

Thank you for this comment.We cited the reference in Reference 3 (please see lines 543-544)

Again, to the reviewer 2, we thank you very much for your kind inputs. We have also learned a lot from your thoughtful comments, which have helped improve this study’s contents.

Sincerely yours,

The Authors

Reviewer 3 Report

Thank you for giving me this opportunity to read the manuscript entitled "An impact asymmetry analysis of small urban green spaces attributes to enhance visitors’ satisfaction". The topic of this manuscript is interesting and would be a good contribution to this field. I think it could be considered for publication in International Journal of Environmental Research and Public Health once the following issues are addressed.

First, there seems to be a typo in the title of the manuscript, namely “… Analysis ff Small …”.

Second, please replace the keywords that already appear in the manuscript’s title with close synonyms or other keywords, which will also facilitate your paper to be searched by potential readers. Besides, I think the first three keywords are a little bit too general.

Third, the fonts of the compass, scale, and legend are so small that it is difficult for the reader to read clearly.

Four, Lines 26-27 – “Urban green spaces with good location and convenient transportation can increase residents’ use (Zhang & Zhou, 2018).”: some newly published papers are suggested to be cited as references here, such as the paper titled "How does urban expansion impact people’s exposure to green environments? A comparative study of 290 Chinese cities". Besides, the paper titled "Observed inequality in urban greenspace exposure in China" is also suggested to be cited in Lines 33-34.

Five, the language of this article is good, but there is still a few grammatical errors. Therefore, a critical review of the manuscript language will improve readability.

Author Response

Manuscript: ijerph-1592446

Title: An impact asymmetry analysis of small urban green spaces attributes to enhance visitors’ satisfaction

Dear Reviewer 3:

The authors would like to thank the area reviewer for your time and comments. Your feedback is very important. We have carefully addressed each of your suggestions and given the corresponding answers.

---------------------------------------------------------------------------------------------------------------------

Reviewer’s Comments

Responses

there seems to be a typo in the title of the manuscript, namely “… Analysis ff Small …”.

Thank you for this comment.We have changed it. 

Second, please replace the keywords that already appear in the manuscript’s title with close synonyms or other keywords, which will also facilitate your paper to be searched by potential readers. Besides, I think the first three keywords are a little bit too general.

Thank you for this comment.We have changed it (please see line 16).The new keywords are “Impact-asymmetry analysis; Nonlinear effect; Park features; Small urban green spaces”.

Third, the fonts of the compass, scale, and legend are so small that it is difficult for the reader to read clearly.

Thank you for this comment. We changed the Figure 1 and Figure 2 to make them more readable.

Four, Lines 26-27 – “Urban green spaces with good location and convenient transportation can increase residents’ use (Zhang & Zhou, 2018).”: some newly published papers are suggested to be cited as references here, such as the paper titled "How does urban expansion impact people’s exposure to green environments? A comparative study of 290 Chinese cities". Besides, the paper titled "Observed inequality in urban greenspace exposure in China" is also suggested to be cited in Lines 33-34.

Thank you for this comment. We cited the two papers in reference 7 and 12 respectively (please see lines 552-553, 564-565).

Five, the language of this article is good, but there is still a few grammatical errors. Therefore, a critical review of the manuscript language will improve readability.

Thank you for this comment. We have checked  the manuscript language.

Again, to the reviewer 3, we thank you very much for your kind inputs. We have also learned a lot from your thoughtful comments, which have helped improve this study’s contents.

Sincerely yours,

 The Authors

Reviewer 4 Report

Dear Authors,

Please find below and attached my comments and suggestions for your work.

Good luck!

Kind regards,

The Reviewer

Review Report Form

Journal: IJERPH (ISSN 1660-4601)

Manuscript ID: ijerph-1592446

Type: Article

Title: An impact asymmetry analysis of small urban green spaces attributes to enhance visitors’ satisfaction

Authors: Pengwei Wang , Bin Zhou * , Lirong Han , Rong Mei

Section: Environmental Earth Science and Medical Geology

Submission Date: 26 January 2022

Dear Authors,

I have carefully analyzed your article entitled “An impact asymmetry analysis of small urban green spaces attributes to enhance visitors’ satisfaction”.

Congratulations for your work and valuable insights reflected in the content of the manuscript!

The structure of my Review Report Form takes into consideration two sections, namely: (A.) General overview of the article and strong points; and (B) Suggestions meant to improve your current manuscript.

(A.) General overview of the article and strong points:

  • General background and aim of the study: According to the authors, urban green spaces have a beneficial effect on the health and well-being of citizens. In continuation, the authors stated their belief according to which understanding the factors influencing visitor satisfaction with urban green spaces contributes to make more informed policies. Also, while referring to the existing literature review in the field, the authors pointed out the fact that prior studies on green spaces satisfaction primarily focused on the linear correlation between small urban green space attributes and satisfaction.
  • Research objectives and methodology used: In terms of methodology used by the authors, it was highlighted that by utilizing the data from visitors in small urban green spaces (SUGS) in Shanghai, China, this study aims to explore the nonlinear influences of SUGS attributes on visitor satisfaction by performing impact asymmetry analysis.
  • Results of the study: Based on the authors’ analysis, it ought to be mentioned that this study classifies the attributes belonging to the small urban green spaces (SUGS) in Shanghai, China, into frustrators, dissatisfiers, hybrids, satisfiers, and delighters. Also, by integrating attribute classification and their performance, the authors found that safety, noise and social interaction are improvement priorities. In continuation, based on the authors’ notes, it should be pointed out that squares and visitors’ behavior should not be ignored in SUGS management, while managers should carefully monitor SUGS attributes of social environment to meet users’ expectations. In the end, the findings of this current study have implications for SUGS management and future research.

(B) Suggestions meant to improve your current manuscript:

Distinguished Authors I would kindly like to suggest the following aspects:

(1.) Closely analyzing the article, since there are some English language improvements and slight corrections that need to be taken care of. Thus, my recommendation would be to carefully proofread the entire manuscript. 

(2.) Also, I have closely analyzed the format of the article, in order to check whether it follows the guidelines which are specific to the publisher. Thus, I have noticed that the current form of your work needs improvement in this regard. So, my kind suggestion is to closely analyze again the guidelines belonging to the publisher, since the article should fit exactly the publisher’s guidelines (for example, while referring to the manner in which you have put the references in the text and at the end of the article). Also, in order to put your article in a better light, I would kindly suggest putting a better emphasis on your research objectives in your abstract.  

(3.) In continuation, the suggestion would also be inserting in your article a few ideas concerning the correlation between effects of the Covid-19 pandemic and the Covid-19 global crisis, sustainability and sustainability assessment, while focusing on the impact asymmetry analysis of small urban green spaces attributes to enhance visitors’ satisfaction. In this context, I had the chance to read a few interesting scientific works recently, among which I would like to mention: An Exploratory Study Based on a Questionnaire Concerning Green and Sustainable Finance, Corporate Social Responsibility, and Performance: Evidence from the Romanian Business Environment. J. Risk Financial Manag. 2019, 12, 162. https://doi.org/10.3390/jrfm12040162; OECD. Green Cities Programme. https://www.oecd.org/regional/greening-cities-regions/46811501.pdf; OECD. Green Growth in Cities. https://www.oecd.org/regional/green-growth-in-cities.htm.

Dear Authors, congratulations once again for your work and valuable insights reflected in the content of the manuscript, and I hope my comments will be of value to you!

Kind regards,

The Reviewer

Author Response

Response to Reviewer 4’s Comments

Manuscript: ijerph-1592446

Title: An impact asymmetry analysis of small urban green spaces attributes to enhance visitors’ satisfaction

Dear Reviewer 4:

The authors would like to thank the area reviewer for your time and comments. Your feedback is very important. We have carefully addressed each of your suggestions and given the corresponding answers.

---------------------------------------------------------------------------------------------------------------------

Reviewer’s Comments

Responses

Closely analyzing the article, since there are some English language improvements and slight corrections that need to be taken care of. Thus, my recommendation would be to carefully proofread the entire manuscript.

We thank the reviewer for your positive feedback and comments. We have proofread the manuscript.

There are too many figures that show the urban parks. Only one or two are enough. Also, only the name and size are enough for each park.

Thank you for this comment.We have deleted the figures in Table 1 and modified the presentation form of Table 1.

Also, I have closely analyzed the format of the article, in order to check whether it follows the guidelines which are specific to the publisher. Thus, I have noticed that the current form of your work needs improvement in this regard. So, my kind suggestion is to closely analyze again the guidelines belonging to the publisher, since the article should fit exactly the publisher’s guidelines (for example, while referring to the manner in which you have put the references in the text and at the end of the article).

Thank you for this comment. We have change the manner in which we put the referecnces in the text and at the end of the article according to publisher’sguidelines.

Also, in order to put your article in a better light, I would kindly suggest putting a better emphasis on your research objectives in your abstract.  

Thank you for this comment.We added the research purpose in the abstract (please see lines 8-10).

In continuation, the suggestion would also be inserting in your article a few ideas concerning the correlation between effects of the Covid-19 pandemic and the Covid-19 global crisis, sustainability and sustainability assessment, while focusing on the impact asymmetry analysis of small urban green spaces attributes to enhance visitors’ satisfaction. In this context, I had the chance to read a few interesting scientific works recently, among which I would like to mention: An Exploratory Study Based on a Questionnaire Concerning Green and Sustainable Finance, Corporate Social Responsibility, and Performance: Evidence from the Romanian Business Environment. J. Risk Financial Manag. 2019, 12, 162. https://doi.org/10.3390/jrfm12040162; OECD. Green Cities Programme. https://www.oecd.org/regional/greening-cities-regions/46811501.pdf; OECD. Green Growth in Cities. https://www.oecd.org/regional/green-growth-in-cities.htm.

Thank you for your recommendation. I have carefully read the articles you recommended. At the same time, I added some related  statements about Covid-19 in the introduction (please see lines 22-23), cited the viewpoint of OECD, explained the relationship between this study and green cities (please see lines 439-442), and cited the article you recommended in the Reference 2 (please see lines 540-542).

Again, to the reviewer 4, we thank you very much for your kind inputs. We have also learned a lot from your thoughtful comments, which have helped improve this study’s contents.

Sincerely yours,

The Authors

Reviewer 5 Report

Dear authors, your manuscript is interesting and valuable.

The Introduction and aim of the study are clear.  The Literature Review is interesting and well-prepared.

The Study Area has no information about natural conditions. In my opinion, it would be also interesting to inform readers about dominant species, natural habitats, existing high conservation value areas, etc. Are there any protected species?  

The Discussion and Conclusions are clearly presented.

Please understand my below comments.

Line 19: “world’ s” - delete a space

Line 23: “contributes to decrease” – the proper version is “contributes to decreasing”

Line 344-347: “It may be due to following two reasons: (i) high−income people can participate in a range of costly activities; (ii) people with high incomes usually have less leisure time, so they have fewer opportunities to use SUGS” –This is a statement, have you got a reference for it? ?  I am not convinced of the accuracy of these two reasons. How do you know that? What if, these people do visit SUGS but in other districts (where there are e.g. more expensive locations for living, more single-family houses )? Maybe they visit SUGS but in other hours, time of the day? What if some people weren’t honest in case of material status.

Line 387: “all the paths in SUGS are relatively curved and narrow” – This is a big simplification. Narrow?, how wide are they (cm.?), are they adapted to wheelchair users?

Line 390-391: “For instance, some SUGS are equipped with sports facilities such as shared basketball courts and badminton courts” – Could you indicate which SUGS have what sports facilities? From Tab. 1, we know only the name, size, and location but nothing about natural conditions or recreation infrastructure.  It could be added to Tab. 1.

Author Response

Response to Reviewer 5’s Comments

Manuscript: ijerph-1592446

Title: An impact asymmetry analysis of small urban green spaces attributes to enhance visitors’ satisfaction

Dear Reviewer 5:

The authors would like to thank the area reviewer for your time and comments. Your feedback is very important. We have carefully addressed each of your suggestions and given the corresponding answers.

--------------------------------------------

Reviewer’s Comments

Responses

The Study Area has no information about natural conditions. In my opinion, it would be also interesting to inform readers about dominant species, natural habitats, existing high conservation value areas, etc. Are there any protected species?  

We thank the reviewer for your comments. The parks located in the center of the city, with a relatively small area. It is mainly planted with trees and shrubs, without high conservation value areas.

Line 19: “world’ s” - delete a space

Thank you for this comment.We have changed it (please see line 19).

Line 23: “contributes to decrease” – the proper version is “contributes to decreasing”

Thank you for this comment.We have changed it (please see line 24).

Line 344-347: “It may be due to following two reasons: (i) high−income people can participate in a range of costly activities; (ii) people with high incomes usually have less leisure time, so they have fewer opportunities to use SUGS” –This is a statement, have you got a reference for it? ?  I am not convinced of the accuracy of these two reasons. How do you know that? What if, these people do visit SUGS but in other districts (where there are e.g. more expensive locations for living, more single-family houses )? Maybe they visit SUGS but in other hours, time of the day? What if some people weren’t honest in case of material status.

Thank you for this comment.We have deleted the two reasons, and changed it to “the reasons for the differences need further research”(please see lines 332-333).

Line 387: “all the paths in SUGS are relatively curved and narrow” – This is a big simplification. Narrow?, how wide are they (cm.?), are they adapted to wheelchair users?

Thank you for this comment. We have changed it (please see lines 406-409). The paths in SUGS are relatively curved and narrow, and their width is about 1-2m. They are suitable for walking and partially suitable for wheelchair users. However, they cannot be used for other activities, such as skateboarding and roller skating.

Line 390-391: “For instance, some SUGS are equipped with sports facilities such as shared basketball courts and badminton courts” – Could you indicate which SUGS have what sports facilities? From Tab. 1, we know only the name, size, and location but nothing about natural conditions or recreation infrastructure.  It could be added to Tab. 1.

Thank you for this comment. We have modified the presentation form of Table 1 and added facilities and natural conditions.

Again, to the reviewer 5, we thank you very much for your kind inputs. We have also learned a lot from your thoughtful comments, which have helped improve this study’s contents.

Sincerely yours,

The Authors

Reviewer 6 Report

Dear Authors,

congratulations on a very interesting and valuable study on the asymmetry analysis of the impact of small urban green areas. The presented research related to the issues of urban green areas, especially small green areas, is part of a wide range of research conducted in this trend. At the same time, the presented results fill a certain gap in the scientific approach to this type of issues by focusing on the analysis of asymmetry, on the basis of which attributes are selected that increase the satisfaction of people visiting small urban green areas.

In my opinion, the authors exhaustively and logically present the way of the research procedure, describing the subsequent stages of their research. The article is well structured, the thoughts and research presented are clearly and substantially correct.

An important aspect of the study is to include in it some limitations that it has. Their awareness and their definition will allow to improve and complete the developed workshop and material.

In the article, I noticed minor imperfections that should be improved to make it more readable and valuable:

  1. Typo in the title - there is "ff" instead of "of" Small Urban Green Spaces
  2. Page 5, Table 1. Details and locations of the nine SUGS - the photos and maps in it are too large in my opinion, while the accompanying legends are too small and illegible, which makes the presentation of individual locations take up 5 pages of the manuscript. I propose to reduce the illustrations and make one common legend for all plans. I think it will be more readable and attractive.
  3. Page 14, Figure 1. The impact of different activities on Satisfaction - the legend of the chart is too small and illegible, I suggest enlarging the markings in it or describing subsequent activities directly on the chart.
  4. Pages 21 and following, References - I noticed errors in the numbering of subsequent items, there are more numbers than the actual items in the literature. There are also errors in editing this part of the article, spaces are missing in many places.

I hope you will find my comments helpful.

Good luck!

Author Response

Response to Reviewer 6’s Comments

Manuscript: ijerph-1592446

Title: An impact asymmetry analysis of small urban green spaces attributes to enhance visitors’ satisfaction

Dear Reviewer 6:

The authors would like to thank the area reviewer for your time and comments. Your feedback is very important. We have carefully addressed each of your suggestions and given the corresponding answers.

---------------------------------------------------------------------------------------------------------------------

Reviewer’s Comments

Responses

Typo in the title - there is "ff" instead of "of" Small Urban Green Spaces

Thank you for this comment.We have changed it.

Page 5, Table 1. Details and locations of the nine SUGS - the photos and maps in it are too large in my opinion, while the accompanying legends are too small and illegible, which makes the presentation of individual locations take up 5 pages of the manuscript. I propose to reduce the illustrations and make one common legend for all plans. I think it will be more readable and attractive.

Thank you for this comment.We modified the presentation form of Table 1, showing more information and making it more readable.

Page 14, Figure 1. The impact of different activities on Satisfaction - the legend of the chart is too small and illegible, I suggest enlarging the markings in it or describing subsequent activities directly on the chart.

Thank you for this comment.We have enlarged the legend of Figure 1.

Pages 21 and following, References - I noticed errors in the numbering of subsequent items, there are more numbers than the actual items in the literature. There are also errors in editing this part of the article, spaces are missing in many places.

Thank you for this comment.We have changed the references.

Again, to the reviewer 6, we thank you very much for your kind inputs. We have also learned a lot from your thoughtful comments, which have helped improve this study’s contents.

Sincerely yours,

 The Authors

Round 2

Reviewer 3 Report

Thank you for giving me this opportunity to read the revised version of the manuscript titled " An impact asymmetry analysis of small urban green spaces attributes to enhance visitors’ satisfaction", and for the detailed responses to my earlier comments. I am satisfied with this revised version, and I think it is acceptable now.